# Antioxidative and Antiglycative Stress Activities of Selenoglutathione Diselenide

**DOI:** 10.3390/ph17081049

**Published:** 2024-08-09

**Authors:** Akiko Kanamori, Nana Egawa, Suyako Yamasaki, Takehito Ikeda, Marcia Juciele da Rocha, Cristiani Folharini Bortolatto, Lucielli Savegnago, César Augusto Brüning, Michio Iwaoka

**Affiliations:** 1Department of Bioengineering, School of Engineering, Tokai University, Kitakaname, Hiratsuka 259-1292, Kanagawa, Japan; n7a0e4w20@outlook.jp (N.E.); denquasuyako@icloud.com (S.Y.); 2Institute of Advanced Biosciences, Tokai University, Kitakaname, Hiratsuka 259-1292, Kanagawa, Japan; 3Department of Chemistry, School of Science, Tokai University, Kitakaname, Hiratsuka 259-1292, Kanagawa, Japan; basketball.boy41@icloud.com; 4Laboratory of Biochemistry and Molecular Neuropharmacology (LABIONEM), Graduate Program in Biochemistry and Bioprospecting (PPGBBio), Chemical, Pharmaceutical and Food Sciences Center (CCQFA), Federal University of Pelotas (UFPel), Pelotas 96010-900, RS, Brazil; marciajr_15@hotmail.com (M.J.d.R.); cbortolatto@gmail.com (C.F.B.); cabruning@yahoo.com.br (C.A.B.); 5Neurobiotechnology Research Group, Graduate Program in Biotechnology, Technologic Development Center, Federal University of Pelotas, (UFPel), Pelotas 96010-900, RS, Brazil; luciellisavegnago@yahoo.com.br

**Keywords:** enzyme mimics, anti-stress agents, oxidative stress, methylglyoxal, cytotoxicity, cell viability

## Abstract

The damage caused by oxidative and glycative stress to cells accumulates on a daily basis and accelerates aging. Glutathione (GSH), a major antioxidant molecule in living organisms, plays a crucial role in detoxifying the stress-causing substances inherent in cells, such as H_2_O_2_ and methylglyoxal (MG), an important intermediate of advanced glycation end-products (AGEs). In this study, we focused on the enhanced antioxidant capacity of the selenium analog of GSH, i.e., selenoglutathione (GSeH), compared to GSH, and examined its effects on the detoxification of stress-causing substances and improvement in cell viability. In cell-free systems, GSeH (1 mM) generated in situ from GSeSeG in the presence of NADPH and glutathione reductase (GR) rapidly reduced more than 80% of 0.1 mM H_2_O_2_, indicating the significant glutathione peroxidase (GPx)-like antioxidant activity of GSeSeG. Similarly, around 50% of 0.5 mM MG was degraded by 0.5 mM GSeH within 30 min through a non-enzymatic mechanism. It was also found that GSeSeG (0.05–0.5 mM) showed glutathione *S*-transferase (GST)-like activity against 1-chloro-2,4-dinitrobenzene (CDNB), a model substance of oxidative stress-causing toxic materials in cells. Meanwhile, HeLa cells that had been pre-treated with GSeSeG exhibited increased viability against 1.2 mM H_2_O_2_ (at [GSeSeG] = 0.5–50 μM) and 4 mM MG (at [GSeSeG] = 3 μM), and the latter effect was maintained for two days. Thus, GSeSeG is a potential antioxidant and antiglycative stress agent for cells.

## 1. Introduction

The damage caused by oxidative and glycative stress to cells accumulates on a daily basis, contributing to the process of aging [1]. Glutathione (GSH) is a major antioxidant molecule in living organisms. It consists of L-glutamic acid, L-cysteine, and glycine (γ-Glu-Cys-Gly). GSH plays a crucial role in detoxifying the stress-causing substances inherent in cells, such as hydrogen peroxide (H_2_O_2_) and methylglyoxal (MG), an important intermediate of advanced glycation end-products (AGEs) that provokes severe glycative stress [2]. GSH utilizes its thiol (SH) group to react with a variety of target molecules within a cell, regulating the redox homeostasis of the cell [3]. The resulting oxidized form of GSH (i.e., GSSG) can be reduced by the enzyme glutathione reductase (GR) back to GSH, which is reused in a reaction with another target molecule [4]. In the cytoplasm, the concentration of GSH is typically maintained in the range of 1–10 mM, and the ratio of [GSSG]/[GSH] is controlled almost constant.

The important roles of GSH in the detoxification of electrophilic harmful compounds in a cell have been well documented. For example, the above-mentioned glutathione redox cycle is a typical protection system against the oxidative stress caused by H_2_O_2_, one of the major reactive oxygen species (ROS) [5]. Furthermore, GSH suppresses the glycative (or carbonyl) stress by reacting with MG in a glyoxalase system [6]. Since MG also inactivates antioxidant enzymes, such as selenium-dependent glutathione peroxidase (GPx), which is involved in the glutathione redox cycle, the accumulation of MG in a cell causes both oxidative and glycative stresses synergistically [7]. Additionally, in a glutathione *S*-transferase (GST) cycle, GSH reacts with various harmful substances to produce GSH conjugates. This is also an important system to protect a cell from poisonous substances [8].

Selenoglutathione (GSeH) is a selenium analog of GSH, having L-selenocysteine (Sec) in place of the Cys residue. GSeH is an essentially non-native peptide. However, Rao and coworkers reported that a small amount of GSeH can be formed when selenium is adopted at high concentrations [9]. This tripeptide exhibits unique biological activities in cell-free systems. GSeH rapidly reacts with H_2_O_2_ to show a higher GPx-like antioxidant activity than the other Sec-containing peptides [10]. It also exhibits a higher radical scavenging activity against various ROS than GSH [11,12]. These high antioxidative activities of GSeH arise from the lower redox potential of GSeSeG/GSeH (−407 mV) compared to GSSG/GSH (−256 mV), indicating that GSeH is a more efficient reductant than GSH [13,14]. Nonetheless, since GSeH is unstable under air atmosphere due to rapid oxidation by oxygen, it is usually isolated in an oxidized diselenide (SeSe) form, i.e., GSeSeG. It should also be noted that GSeSeG can be reduced to GSeH by GR (Figure 1) [13]. GSeSeG is also a useful catalyst for the oxidative folding of various SS-containing proteins [13,15,16,17]. However, little is known about the biological activities of GSeH (or GSeSeG) within living cells, except for the report by Hilvert and coworkers, who applied GSeSeG to in vivo protein folding using *E. coli* cells [18].

In this study, we focused on the high antioxidant capacity of GSeH. We investigated its effects on the detoxification of stress-causing substances in cell-free systems, including H_2_O_2_, MG, and 1-chloro-2,4-dinitrobenzene (CDNB), a model substance for oxidative stress-causing toxic materials in cells. We also assayed the viability of HeLa cells pre-treated with a low concentration of GSeSeG in the presence of H_2_O_2_ and MG. The results suggested that the GSeSeG/GSeH system has the potential to exhibit antioxidant and antiglycative stress capacities not only in vitro but also in vivo.

## 2. Results and Discussion

### 2.1. GPx-like Antioxidant Activity of GSeSeG

Selenium is an essential micronutrient for animals as a source of selenocysteine (Sec), which is incorporated into the active site of several enzymes, such as GPx. GPx is an antioxidant enzyme that catalyzes the reduction of hydroperoxides utilizing GSH [2,19]. The enzymatic reactions of GPx have been well studied, and numerous GPx mimics have already been developed for potential applications as antioxidant drugs [20,21,22,23,24,25,26]. Previously, the GPx-like activity of GSeSeG was assayed in the presence of H_2_O_2_, NADPH, and GR. The reduction rate of H_2_O_2_ by GSeSeG was indirectly measured by monitoring the decrease in the UV absorbance of NADPH at 340 nm [10]. In contrast, the present study directly measured the reduction rate of H_2_O_2_ by employing the FOX assay, a common method for determining H_2_O_2_ concentration in a sample solution [27]. Here, GSeH was first generated from GSeSeG in the presence of NADPH and GR (see Appendix A). The generated GSeH (ca. 89% conversion) was then reacted with H_2_O_2_. Figure 1 shows the residual rates of H_2_O_2_ under the various assay conditions.

GSeH (1 mM) showed high rates of H_2_O_2_ reduction (>80%) regardless of the presence of GPx (+GPx) and GSH (+2 mM GSH), which was added to the assay solution to mimic the cytosolic condition. The results indicated that GSeH reduced H_2_O_2_ in a non-enzymatic manner. In contrast, the rates of H_2_O_2_ reduction by GSH (1 mM) as a reference depended on both GPx (0.014 units/mL) and additional GSH (2 mM) coexisting in the assay solution. In the presence of GPx (+GPx), the reduction of H_2_O_2_ with GSH should be catalyzed by the enzyme because the H_2_O_2_ reduction rate became low in the absence of GPx, where H_2_O_2_ should be reduced through a direct reaction with GSH. The decrease in the rate of residual H_2_O_2_ from around 60 to 40% by addition of GSH (+2 mM GSH), observed in the absence of GPx, supported the occurrence of a direct reaction between H_2_O_2_ and GSH. By comparing the rates of the unreacted H_2_O_2_, it is clear that GSeH is a stronger reductant than GSH. This is in accord with the dramatic activity loss of GPx by the Sec-to-Cys mutation at the active site [28]. It should also be noted that the range of the GSH concentrations applied here is at the lower limit or within a normal range of [GSH] in the cytoplasm, i.e., 1.0–10 mM. The results of Figure 1 clearly reconfirmed that GSeSeG has GPx-like antioxidant activity in the presence of NADPH and GR [10].

### 2.2. Glyoxalase 1 (GLO 1)-like Antiglycative Stress Activity of GSeSeG

GLO 1 is an enzyme that catalyzes the degradation of harmful MG into *S*-lactoylglutathione (MeCH(OH)CO-SG) using GSH [29,30]. The reaction of MG with GSH first generates a hemithioacetal intermediate, which is subsequently converted into *S*-lactoylglutathione by the function of GLO 1. This study assayed the GLO 1-like activities of GSeH and GSH (as a reference), which were generated in situ from GSeSeG and GSSG, respectively, in the presence of NADPH and GR. The assay measured the rates of unreacted MG in the reaction solutions of MG (0.5 mM) and GSeH or GSH (0.5 mM). To achieve this, the unreacted MG was reacted with 1,2-diamino-4,5-methylenedioxybenzene dihydrochloride (MDB), and the amounts of the generated MG–MDB adduct were determined by the fluorescence of the solution at 393 nm (Figure 2) [31]. The results clearly show that MG was rapidly consumed in the presence of GSeH (around 50% in 30 min), while the reaction between GSH and MG was slow: the rates of unreacted MG were 97, 96, 87, and 60% after 0.5, 2, 5, 24 h, respectively. However, the rates were around 35% after 48 h for both the reactions of MG with GSeH and GSH. In the HPLC analysis of the reaction solution of GSeH, only one peak, corresponding to the MG–MDB adduct, was detected by the fluorescence at 393 nm (Appendix A). The peak intensity decreased in consonance with the residual MG rates observed in Figure 2. Thus, the GLO 1-like antiglycative stress activity of GSeSeG in the presence of GR and NADPH was confirmed, although the reaction products between GSeH and MG could not be identified in the assay system.

The above cell-free experiments (Figure 1 and Figure 2) demonstrated that GSeSeG can exert anti-stress activities through GPx-like and the GLO 1-like mechanisms. Therefore, GSeSeG is an effective mimic of GPx and GLO 1 enzymes and could be applied as a pre-catalyst to suppress oxidative and glycative stresses in cells.

### 2.3. Glutathione S-Transferase-like Activity of GSeSeG

GST is a designation for enzymes that catalyze the reaction of GSH with intracellular toxins and drugs that have electrophilic properties. GSTs are essential for detoxifying these xenobiotics (X) in vivo into glutathione conjugates (GS-X), which are transported outside the cells and eventually from the body [8]. Using 1-chloro-2,4-dinitrobenzene (CDNB) as a model substance (X), the rate of the reaction between GSH (2 mM) and CDNB (1 mM) in the presence of GST (0.15 units/mL) was compared with that in the presence of GSeSeG (0.05 to 0.5 mM) (Figure 3).

The catalytic activities of GST and GSeSeG were determined by the change in absorbance at a wavelength of 340 nm, which corresponds to the GSH conjugates of CDNB (GS-DNB) (Figure 3A). Although the activity of GSeSeG was lower than GST, GSeSeG showed significant GST-like activity compared to the control at 0.5 mM, the same concentration used in the GLO 1-like antiglycative stress assay (Figure 2). Correspondingly, the GST-like activity decreased with lower concentrations of GSeSeG. It is notable that even at the lowest applied concentration ([GSeSeG] = 0.05 mM), GSeSeG still exhibited remarkable GST-like activity in this assay.

We subsequently analyzed the reaction products by HPLC (Figure 3B,C). According to the literature [32,33], the products were detected at 246 nm, corresponding to the absorption of CDNB, as well as at 340 nm, corresponding to the absorption of the GS-DNB adduct. In the presence of GST, only one peak (indicated as peak 1 in Figure 3B) was detected at 340 nm, which eluted at around 9 min of the retention time. In contrast, two peaks were eluted in the presence of GSeSeG at around 9 and 12 min of the retention times (indicated as peaks 1 and 2, respectively, in Figure 3C). These peaks were fractionated and analyzed by ESI(+)-TOF-MS (Figure 3D,E), which confirmed peak 1 as GS-DNB and peak 2 as a conjugate of GSeH and CDNB (GSe-DNB). Thus, it was found that in the presence of GSeSeG both GS-DNB and GSe-DNB were formed, while only GS-DNB was generated in the presence of GST. Interestingly, the formation of the same GS-DNB was detected in the control solution, which did not contain GST, indicating that GSH can slowly react with CDNB through a non-enzymatic path. Similarly, GSeH was capable of reacting with CDNB directly, whereas GSeSeG did not react with CDNB. Thus, it was strongly suggested that GSeH was generated from GSeSeG by reacting with GSH and the generated GSeH reacted with CDNB to produce GSe-DNB. The existence of this non-enzymatic process for the conversion of GSeSeG into GSeH, in addition to the GR-catalyzed enzymatic process (Figure 1), is a crucial characteristic of GSeSeG for its application in in vivo studies, considering that GSH is ubiquitously present in living organisms at high concentrations.

### 2.4. Effects of GSeSeG on Cell Viability against Oxidative Stress Caused by H_2_O_2_

The antioxidative and antiglycative stress activities of GSeSeG have been observed in cell-free systems, as previously mentioned. To assess these activities in living cells, we subsequently carried out cell viability tests. HeLa cells were chosen for this preliminary study due to their adherent nature and ease of use in assays under various conditions. The cells were treated with GSeSeG (25 or 50 μM) in advance and then exposed to oxidative stress by the addition of H_2_O_2_ (1 mM). The concentrations of GSeSeG and H_2_O_2_ were pre-adjusted so that the cell viability remained around 50%. During the pre-incubation, GSeSeG was imported into the cells and reduced by NADPH and GR to GSeH, which then decreased the oxidative stress caused by H_2_O_2_ in the cytoplasm through the antioxidative cycle. The results are shown in Figure 4.

Without the addition of H_2_O_2_, the cell viability remained constant (around 100%) with the treatment of GSeSeG (25 or 50 μM). This result indicated that GSeSeG is non-toxic at a concentration of at least up to 50 µM. This property of GSeSeG was remarkable as compared with the cytotoxicity of selenite (Na_2_SeO_3_), for which the LB50 value was 1.2 μM against HeLa cells [34]. Similarly, GSH and GSSG as references did not show any toxic effects at concentrations of 100 and 50 µM, respectively. However, when H_2_O_2_ (1 mM) was added, GSH and GSSG at the same concentrations showed a significant decrease in cell viability (44 and 33%, respectively). No improvement in cell viability was observed compared to the control (41%). Since GSH and GSSG are already present at high concentrations (>1 mM) in the cytoplasm, these results suggest that the oxidative stress caused by 1 mM H_2_O_2_ was too strong to be mitigated by the normal concentration levels of GSH and GSSG. In contrast, GSeSeG at the concentrations of 25 and 50 µM improved the cell viability against H_2_O_2_ stress to over 80% (83 and 88%, respectively). To assess the antioxidant capacity of GSeSeG more quantitatively, HeLa cells’ viability against a slightly increased oxidative stress by 1.2 mM H_2_O_2_ was tested in the presence of varying GSeSeG concentrations (Figure 4B). In this experiment, we used a slightly higher concentration of H_2_O_2_ (1.2 mM) because the cell viability against 1 mM H_2_O_2_ stress was a little too high (more than 80%) to analyze the concentration-dependent effects of GSeSeG. Surprisingly, pre-treatment addition of GSeSeG at even 0.5 µM improved cell resistance to oxidative stress from 21 to 57%. This definitively proves the high antioxidant capacity of GSeSeG in cells. Although the mechanism of the GSeSeG incorporation through the cell membrane is unknown [18], the results of this cell viability test strongly support its ability to enter cells and exhibit antioxidative stress activity in the cytoplasm.

GSeSeG, like GSSG, exhibits high water solubility. Therefore, this small tripeptide dimer is expected to possess a low toxicity in vivo [34]. The results in Figure 4A support this notion, as GSeSeG showed no toxicity to HeLa cells at the concentrations of at least up to 50 µM. This concentration is significantly higher than those required for GSeSeG to exert its antioxidative stress activity in the cells (Figure 4B). Therefore, the cytotoxicity of GSeSeG would be overcome by controlling the concentration when it is applied in vivo.

### 2.5. Effects of GSeSeG on Cell Viability against Glycative Stress Caused by MG and the Continuation of the Effects

The addition of MG to cells causes glycative (or carbonyl) stress and eventually induces cell death [35]. To assess the effects of GSeSeG on MG-induced stresses, HeLa cells were first grown to a uniform subconfluent density. Then, GSeSeG (3.0 µM) was added to the medium and incubated overnight to incorporate GSeSeG into the cells. On the second day, the medium was replaced with fresh medium containing 0 or 4 mM MG and incubated overnight. On the third day, the medium was replaced with fresh medium containing H_2_O_2_ (0 or 1.2 mM) and incubated overnight. The viability of the cells was examined after replacing the medium with fresh medium on the fourth day. The H_2_O_2_ concentration added on the third day was set to the same as that investigated in Figure 4B.

The effects of adding 3 µM GSeSeG on cell viability against the glycative stress caused by 4.0 mM MG are shown in Figure 5. It is seen that the addition of MG significantly decreased cell viability, depending on the concentration of H_2_O_2_ subsequently added to the medium. By comparing the results obtained in the presence and in the absence of GSeSeG, it is clear that GSeSeG improved the cell viability against glycative as well as oxidative stress. Without a treatment of H_2_O_2_, the glycative stress caused by 4 mM MG was rescued by the pre-treatment with 3 µM GSeSeG: the ratio of the living cells was increased from 38 to 56% on the average. However, the increment was statistically not significant due to large errors. On the other hand, when 1.2 mM H_2_O_2_ was post-loaded, the ratio was significantly increased from 26 to 38% by the pre-treatment with 3 µM GSeSeG. The antioxidative stress activity of GSeSeG was also observed in the presence of 1.2 mM H_2_O_2_ without the addition of 4 mM MG.

It should be noted that the effects of GSeSeG on cell viability continued for at least two days because GSeSeG was added to the medium one day before the addition of MG and two days before the addition of H_2_O_2_. Thus, not only the production of GSeH but also an efficient recycling mechanism for GSeSeG should be working in the cytoplasm.

As shown in Figure 1, active GSeH can be generated by the enzymatic function of GR, which is abundant in cells along with NADPH. This enzymatic pathway should be readily functional in cells once GSeSeG is imported into the cytoplasm. In addition to this enzymatic mechanism, the presence of another effective path to activate GSeSeG into GSeH was identified in this study: as shown in Figure 3, the GST-like activity of GSeSeG was observed in the presence of excess GSH, indicating that GSeH can also be generated from GSeSeG non-enzymatically. The presence of this non-enzymatic pathway is of importance when GSeSeG is applied in vivo because GSH is ubiquitously present in high concentrations within cells. Therefore, during the cell viability assays with H_2_O_2_ and/or MG (Figure 4 and Figure 5), both enzymatic and non-enzymatic pathways likely contributed to the activation of GSeSeG to GSeH.

### 2.6. Detoxification Functions of GSeSeG in Cells

A plausible mechanism for the detoxification of stress-causing materials, such as H_2_O_2_, MG, CDNB, etc., in cells is summarized in Figure 6. Although the mechanisms for the incorporation of GSeSeG into cells and the efflux of the GSeH conjugates as end-products outside the cells are not clear, GSeSeG degrades the stress-causing materials in the activated GSeH form, which is generated either through an enzymatic or non-enzymatic pathway. The reaction of GSeH with H_2_O_2_ regenerates GSeSeG, while the reactions with MG and CDNB generate the GSeH conjugates through the non-enzymatic path. Thus, the plausible mechanism of GSeSeG involves several non-enzymatic reactions, which is in significant contrast to the reactions of the sulfur counterpart, i.e., GSSG [2,3,4]. In the case of GSSG, the activation to GSH is catalyzed by GR, and the reactions with MG and CDNB are mostly catalyzed by GLO 1 and GST, respectively. The difference is due to the higher redox reactivity of selenium than that of sulfur. Thus, GSeSeG has unique features as an anti-stress agent and would have advantages in biological applications.

## 3. Materials and Methods

### 3.1. Chemicals and Reagents

Glutathione peroxidase (GPx) from bovine erythrocytes, glutathione *S*-transferase (GST), and 1-chloro-2,4-dinitrobenzene (CDNB) were purchased from Sigma–Aldrich (St. Louis, MO, USA). Trypsin-ethylenediaminetetraacetic acid (EDTA) solution (containing 0.05% trypsin), fetal bovine serum (FBS), Dulbecco’s Modified Eagle’s Medium (DMEM) with a high glucose content, and Opti-MEM I reduced serum medium were from Thermo Fisher Scientific, Life Technologies Ltd., under the brand Gibco (Waltham, MA, USA). Reduced and oxidized glutathione (GSH and GSSG), glutathione reductase (GR) from yeast, nicotinamide adenine dinucleotide phosphate reduced form (NADPH), hydrogen peroxide (H_2_O_2_) 30% (*w*/*w*), methylglyoxal (MG) 40% (*w*/*w*), ethylenediaminetetraacetic acid (EDTA), and neutral red (NR) were from Fujifilm Wako Pure Chemical Corporation (Tokyo, Japan). Unless otherwise indicated, all other guaranteed reagents were also from Fujifilm Wako Pure Chemical Corporation (Tokyo, Japan).

1,2-Diamino-4,5-methylenedioxybenzene dihydrochloride (MDB) was from Dojindo Laboratories (Kumamoto, Japan). Solvents with a minimum of HPLC-grade purity were from Kanto Chemical Corporation (Tokyo, Japan). Selenoglutathione diselenide (GSeSeG) was synthesized according to our previous method (Appendix A) [17]. The purity and identity of the synthesized GSeSeG were confirmed by the RP-HPLC analysis (see Appendix A). A phosphate-buffered saline at pH 7.4 (PBS) used in the preparation of the assay solutions was degassed thoroughly with argon gas in advance.

### 3.2. Preparation of 1 mM GSeH in PBS from GSeSeG by Treatment with GR and NADPH

Solutions of 1 mM GSeH in PBS, which were used in the GPx-like antioxidant and the glyoxalase 1 (GLO 1)-like antiglycative stress assays (vide infra), were prepared from GSeSeG in the presence of GR and NADPH according to the slightly modified procedure reported previously [10]. Briefly, a 0.5 mM GSeSeG solution in PBS was reacted with NADPH (1.5 mM) and GR (10 units/mL) for 10 min at 37 °C either in the presence or absence of 2 mM GSH. For comparison, 1 mM GSH solutions in PBS were also prepared from GSSG using GR and NADPH in the presence or absence of 2 mM GSH. This resulted in four test solutions: 1 mM GSeH w/2 mM GSH, 1 mM GSeH w/o GSH, 1 mM GSH w/additional 2 mM GSH, and 1 mM GSH w/o additional GSH.

### 3.3. Determination of Glutathione Peroxidase (GPx)-like Antioxidant Activity

The GPx-like antioxidant activities (i.e., H_2_O_2_ degradation activities) of the prepared test solutions were assayed as follows. A test solution prepared in PBS (100 μL) as described above was added with 0.438 units/mL GPx (3.3 μL) and 3 mM H_2_O_2_ (3.3 μL) solutions to make an assay solution, in which the concentrations of GPx and H_2_O_2_ were 0.014 units/mL and 0.1 mM, respectively. After incubation at 37 °C for 30 min, the concentration of remaining H_2_O_2_ in the assay solution was determined by the FOX assay according to the procedure reported in the literature with slight modifications [27]. Briefly, a portion of the solution (20 µL) was mixed with 180 µL of the FOX working reagent containing 0.12 mM xylenol orange, 0.25 mM Fe(NH_4_)_2_(SO_4_)_2_, and 25 mM H_2_SO_4_. The mixture was maintained at room temperature for 20 min in a 96-well plate. The concentration of H_2_O_2_ was monitored by the absorbance at 595 nm using a microplate reader (SH-9000, Corona Electric Co., Ltd., Ibaraki, Japan). Similar assays were performed without the addition of the GPx solution. As a control, PBS was employed instead of the test solution. The assays were run in triplicate.

### 3.4. Determination of Glyoxalase 1 (GLO 1)-like Antiglycative Stress Activity

The GLO 1-like activities (i.e., MG degradation activities) of the prepared test solutions were assayed according to the procedure supplied by the manufacturer with slight modifications. A test solution (300 μL) was added to a 1.0 mM MG solution (300 μL), which was prepared by diluting the MG solution 40% (*w*/*w*) with PBS. After incubation at 37 °C for 0.5, 2, 5, 24, and 48 h, an aliquot (100 µL) of the assay solution ([GSeH or GSH] = [MG] = 0.5 mM) was taken and added to a 1 mM MDB solution (100 µL) to quench the reaction. After incubation at 60 °C for 40 min, a portion of the mixture (100 µL) was transferred into a 96-well plate. The amount of unreacted MG (i.e., the amount of MG–MDB adduct) was determined by fluorescence with excitation and emission wavelengths at 355 and 393 nm, respectively, using a microplate reader (SH-9000, Corona Electric Co., Ltd., Ibaraki, Japan). In the meantime, the products in the reaction solution were analyzed by RP-HPLC using a Tosoh ODS-100V (4.6 mm × 150 mm, Tosoh Corporation, Tokyo, Japan) column at 35 °C at a flow rate of 0.7 mL/min. An isocratic solvent (MeOH:acetonitrile:40 mM phosphate buffer at pH 7.0 = 44:7:49) was applied. The reactions were run five times.

### 3.5. Determination of Glutathione S-Transferase (GST)-like Activity

The GST-like activity of GSeSeG was estimated by the formation of the GSH–CDNB conjugate, which has an absorption at 340 nm. According to the literature procedure [36,37], all the reagents were prepared in a 0.1 M sodium phosphate buffer solution at pH 6.9, except for CDNB, which was dissolved in ethanol instead of the buffer solution. Briefly, a 4 mM GSH solution (200 μL) was mixed with a sample solution (i.e., 0.6 unis/mL GST or 1.2 mM GSeSeG) (100 µL) at 37 °C for 5 min. To the mixture, a 4 mM CDNB solution (100 µL) was added to make an assay solution (a total volume of 400 µL) containing 2 mM GSH, 0.15 units/mL GST or 0.3 mM GSeSeG, and 1 mM CDNB. Similarly, the assay solutions containing 2 mM GSH, 0.05, 0.1, 0.2, or 0.5 mM GSeSeG, and 1 mM CDNB was prepared.

A portion of the assay solution (100 µL) was dispensed in a 96-well plate and incubated at 37 °C for 10 min. The absorbance at 340 nm was measured using a microplate reader (SH-9000, Corona Electric Co., Ltd., Ibaraki, Japan) to estimate the amounts of GSH–CDNB conjugate formed in the assay solution. The GST-like activity of the sample was defined as the slope for the plots of the absorbance at 340 nm as a function of the reaction time (Δ*A*_340_/min). The measurements were run in triplicate.

To characterize the molecular structure of the reaction products, the solutions obtained 60 min after the reaction were analyzed by RP-HPLC using an InertSustain AQ-C18 (4.6 mm × 150 mm, GL Science Inc., Tokyo, Japan) column at 35 °C at a flow rate of 1.0 mL/min. An isocratic solvent (35%B; solvent A, 0.1% formic acid in H_2_O; solvent B, 0.1% formic acid in methanol) was applied for the initial 10 min, and then %B was linearly increased from 35 to 100% in 7 min. Detection wavelengths were set to 246 and 340 nm to monitor all substances and only CDNB conjugates, respectively. The detected CDNB conjugates were collected and analyzed by ESI(+)-TOF-MS on a JMS-T100LP mass spectrometer (JEOL Ltd., Akishima, Japan) in a high-resolution mode.

### 3.6. Cell Cultures

HeLa cells were purchased from the National Institute of Health Science (Kawasaki, Japan). DMEM (500 mL) was supplemented with 10% FBS without antibiotics (55 mL). In the DMEM solution, HeLa cells were cultured at 37 °C under a 5% CO_2_ atmosphere. The cells were then seeded into a 96-well plate at an initial cell density of approximately 3 × 10^4^ cells per cm^3^ and incubated overnight for subsequent viability assays.

### 3.7. Cell Viability Assays

The viability of the cultured HeLa cells against H_2_O_2_ and/or MG was investigated in the presence of GSH, GSSG, or GSeSeG.

For the assays against H_2_O_2_, HeLa cells in the subconfluent state in a 96-well plate were washed with Opti-MEM (100 μL) once, followed by the addition of a test substance (i.e., GSH, GSSG, or GSeSeG) dissolved in Opti-MEM (100 μL) at various concentrations ([GSH] = 100 μM, [GSSG] = 50 μM, [GSeSeG] = 0 to 50 μM). After incubation of the cells overnight, the medium was removed by aspiration. To the obtained cells was then added a 1.0 or 1.2 mM H_2_O_2_ solution in Opti-MEM (100 μL). The cells were incubated overnight.

For the assays against MG, HeLa cells in the subconfluent state in a 96-well plate were washed with Opti-MEM (100 μL) once, followed by the addition of a 3 µM GSeSeG solution in Opti-MEM (100 μL), and then incubated overnight. After aspiration of the medium, a 0 or 4.0 mM MG solution in Opti-MEM (100 μL) was added to the cells. The cells were incubated overnight. After removal of the MG-containing medium, the cells were treated with a 0 or 1.2 mM H_2_O_2_ solution in Opti-MEM (100 μL). The mixture was incubated overnight.

The ratios of the living HeLa cells were then determined by the degree of incorporation of NR into the cells following the method of Lillig et al. [38], with slight modifications. Briefly, after checking the cells’ growth under a microscope, the cell layers were incubated for 2–3 h in a 40 µg/mL NR solution in Opti-MEM (100 μL) to allow dye accumulation. Then, the cells were fixed using an isotonic phosphate buffer solution at pH 7.4 containing 1% paraformaldehyde (200 μL). After the lysis of the fixed cells using 50% ethanol containing 1% acetic acid (100 μL), the resulting lysates were analyzed spectrophotometrically by the absorbance at 540 nm using a microplate reader (SH-9000, Corona Electric Co., Ltd., Ibaraki, Japan). The cells untreated with H_2_O_2_ or MG but with NR served as a 100% standard, while the cells untreated with NR served as a 0% standard. The assays were run in more than triplicate.

### 3.8. Statistical Analysis

The values were expressed using means and standard deviations (SDs). The one-way analysis of variance (ANOVA) was used to determine whether there were any statistically significant differences among the means of independent groups. A Newman–Keuls test was performed as a post hoc analysis to address the pairs of groups that had a significant difference. The *t*-test was used to examine the comparisons between two groups. When *p* < 0.05, the differences were regarded as statistically significant. All statistical analyses were carried out by the Microsoft Excel program using built-in functions for the one-way ANOVA and *t*-test. For the post hoc test, the analysis was manually performed using the same program.

## 4. Conclusions

In this study, we investigated the antioxidative and antiglycative stress activities of GSeSeG, both in cell-free systems and in cells. As a result, several advantageous features of GSeSeG as an anti-stress agent became evident. First, GSeSeG has GPx-like antioxidant activity against H_2_O_2_. Second, GSeSeG exhibits GLO 1-like antiglycative stress activity against MG. Third, the GST-like detoxification activity of GSeSeG against CDNB was demonstrated. The antioxidative and antiglycative stress activities of GSeSeG were also confirmed by the viability assays using HeLa cells, as the cell viability was recovered from the H_2_O_2_-induced oxidative stress by pre-treatment of the cells with GSeSeG (Figure 4). Similarly, the pre-treatment with GSeSeG rescued the cells from the MG-induced glycative (or carbonyl) stress to a significant extent (Figure 5). Moreover, the cytotoxicity of GSeSeG was found to be low (i.e., non-toxic at concentrations up to 50 μM). This is an advantage of GSeSeG as a potential anti-stress selenium agent, because selenium compounds are known to be highly toxic [34]. It was also found in this study that the active form of GSeSeG, i.e., GSeH, can be generated in both cell-free systems and in cells through not only enzymatic (by the function of GR) but also non-enzymatic paths (by the reaction with GSH). Thus, GSeSeG could be a potential anti-stress agent for various biological applications. However, since GSeSeG is a small peptide, it would be hydrolyzed rapidly in the body. Therefore, the enhancement of biostability as well as the rapid delivery to the target cells or organisms should be the issue of future study.

## Data Availability

The original contributions presented in the study are included in the article and Appendix A. Further inquiries can be directed to the corresponding authors.

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
