# Peer review of "Antioxidative and Antiglycative Stress Activities of Selenoglutathione Diselenide"

_pharmaceuticals, 2024, doi:10.3390/ph17081049_

Round 1
Reviewer 1 Report
Comments and Suggestions for Authors
The manuscript entitled "Antioxidative and antiglycative stress activities of selenoglutathione diselenide" describe the antiglycative and antioxidative potential of selenium derivative of glutathione in which sulfur is replaced with selenium. The studies are designed well and explained scientifically in organized manner. The manuscript is acceptable after minor corrections.
1. The synthetic methodology by which selenoglutathione diselenide is prepared should be discussed briefly or should be submitted as supplementary information. The characterization of synthesized compound with spectra data is key for determination the purity of final tested compound.
2. The discussion part can be more suitable along with results for better understanding of readers. I would suggest to include this part along with results under section results and discussion.
3. The significance of results is needed to be added for comparison of p value of each other.
4. The discussion part along with results also need to be compared to literature reports for proof of plausible mechanism.
5. The manuscript need to be revised for grammatical and typographical errors.
Author Response
Comments 1: The synthetic methodology by which selenoglutathione diselenide is prepared should be discussed briefly or should be submitted as supplementary information. The characterization of synthesized compound with spectra data is key for determination the purity of final tested compound.
Response 1: GSeSeG was synthesized according to our previous method [ref. 17]. This is now clearly mentioned in the subsection, 3.1. Chemicals and reagents. The brief protocol is shown in new Supporting Materials. We agree that the characterization of GSeSeG is essential for the reliability of the assay data. Therefore, we show the HPLC chromatograms of GSeSeG and GSeH, which was generated from GSeSeG by NADPH and GR, in new Supporting Materials as Figures S1 and S2, which clearly confirmed the purity of GSeSeG and the efficient conversion (ca. 89 %) of GSeSeG to GSeH by NADPH and GR.
Comments 2: The discussion part can be more suitable along with results for better understanding of readers. I would suggest to include this part along with results under section results and discussion.
Response 2: According to the suggestion, Discussion section is now merged to Results section as Results and Discussion.
Comments 3: The significance of results is needed to be added for comparison of p value of each other.
Response 3: We analyzed the p values for all the data shown in Figures 1 to 5, and the results are now included in Figures 1, 4, and 5. However, it seems to be trivial to show p values for Figures 2 and 3. Therefore, the p values are not included for these figures. Please note that Figure 5 has been reformatted for the sake of clear understanding.
Comments 4: The discussion part along with results also need to be compared to literature reports for proof of plausible mechanism.
Response 4: We revised the Results and Discussion sections largely according to the suggestions by three reviewers. We believe that the revised manuscript is now much clearer than the original one and the plausible mechanism shown in Figure 6 becomes trustworthy.
Comments 5: The manuscript need to be revised for grammatical and typographical errors.
Response 5: We made our best efforts to correct grammatical and typographical errors.
Reviewer 2 Report
Comments and Suggestions for Authors
The submitted manuscript describes the high antioxidant capacity of selenoglutathione (GSeH) and the investigation of GSeH’s effects on detoxification of the stress-causing substances using cell-free systems. Although the experimental design shown in the submitted manuscript seems to be fine, the explanation of experimental results is critically bad. All graphs should be created accurately, and the contents of the graph must be explained carefully. All Figure legends should be rebuilt. It is too hard to understand the contents of Figure (graph) from each legend. Based on these points, the reviewer judges that the submitted manuscript should be revised largely for publication for “pharmaceuticals”.
Figure 1 should be rebuilt. The reviewer does not understand why GSeH is bold. Also, the reviewer does not understand the paragraph from line 106 to 118. What is “additional GSH”? (line 113 and 114) The explanation of Figure 2A is not sufficient. The authors should show the raw data on Figure 2B. The explanation of Figure 3A is strange. The authors should explain why the authors observe the data at 246 nm. There is no basis on the concentrations (0.3 mM) of GST and GSeSeG in the figure legend of Figure 3B and 3C. The reviewer does not understand the sentences from line 174 to 178. The authors should show why the authors select indicated GSeSeG concentrations in Figure 4A and 4B. Also, the authors should show why the authors select indicated H2O2 concentrations in Figure 4A and 4B. Why do the authors use the different H2O2 concentration in this experiment? The authors should explain the difference of the viability of the control experiments conducted in the presence of H2O2. The sentence of “GSeSeG even at the concentration of 25 µM improved the cell viability against the H2O2 stress” (line 204 to 205) is strange. The reviewer does not understand the sentences from line 208 to 209. The authors should explain why H2O2 concentrations (0, 1.2, 1.5 mM) are selected. The reason in described in line 228 is not persuasive. Figure 5A and 5B indicate reduction of the cell viability depending on H2O2 concentration. Therefore, the reviewer does not understand why the authors describe “the addition of MG significantly decreased the cell viability irrespective of the concentration of H2O2” (line 231). The expression described in line 232 is strange. The authors should explain the experimental results shown in Figure 5, accurately and briefly. Subchapters in Discussion is unnecessary. If the authors would like to separate the contents, the authors should use the paragraph. The reviewer does not understand the contents of “3.3. Low cytotoxicity of GSeSeG”. The reviewer thinks that the contents make it impossible to discuss the low toxicity on GSeSeG. Figure 6 should be rebuilt. “GSeSeG” (placed to the outer side of the cell) should be linked to “GSeSeG” in the cell, and “GSe-X” (placed to the outer side of the cell) should be linked to “GSe-X” in the cell.
Comments on the Quality of English LanguageModerate English proofreading is required.
Author Response
Comments 1: The submitted manuscript describes the high antioxidant capacity of selenoglutathione (GSeH) and the investigation of GSeH’s effects on detoxification of the stress-causing substances using cell-free systems. Although the experimental design shown in the submitted manuscript seems to be fine, the explanation of experimental results is critically bad. All graphs should be created accurately, and the contents of the graph must be explained carefully. All Figure legends should be rebuilt. It is too hard to understand the contents of Figure (graph) from each legend. Based on these points, the reviewer judges that the submitted manuscript should be revised largely for publication for “pharmaceuticals”.
Response 1: Thanks for the critical and valuable comments. We took all the comments raised by this reviewer seriously and have revised all the figures and the legends largely as replied below.
Comments 2: Figure 1 should be rebuilt. The reviewer does not understand why GSeH is bold. Also, the reviewer does not understand the paragraph from line 106 to 118. What is “additional GSH”? (line 113 and 114)
Response 2: Figure 1 was rebuilt as suggested. In this experiment, we used GSH (1 mM) as a reference to GSeH (1 mM). Therefore, when 2 mM GSH was added to the assay solution, the total concentration of GSH became 3 mM. To make this point clear, the caption of Figure 1 and the following paragraph were revised. According to these revisions, we believe that the meaning of additional GSH is clear now.
Comments 3: The explanation of Figure 2A is not sufficient. The authors should show the raw data on Figure 2B.
Response 3: The explanation of Figure 2A, provided in the first paragraph of subsection, 2.2. Glyoxalase 1 (GLO 1)-like antiglycative stress activity of GSeSeG (p.4), is appended. The raw data of the HPLC chromatograms shown in Figure 2B are now provided in Supplementary Materials as Figure S3.
Comments 4: The explanation of Figure 3A is strange. The authors should explain why the authors observe the data at 246 nm. There is no basis on the concentrations (0.3 mM) of GST and GSeSeG in the figure legend of Figure 3B and 3C. The reviewer does not understand the sentences from line 174 to 178.
Response 4: Explanation of Figure 3A was slightly modified (p.5). In the HPLC analysis for the products of the reaction between GSH and CDNB, we used two wavelengths, i.e., 246 and 340 nm, according to the literature [new references 32 and 33]. The absorption at 246 nm was used for detecting CDNB, while the absorption at 340 nm was used for detecting GS-DNB. In the figure legend, we added the concentration of GST, which was missing in the original manuscript. We revised the relevant part (l.190-195) to make the point clear that GSeH can be generated from GSeSeG by the reaction with GSH.
Comments 5: The authors should show why the authors select indicated GSeSeG concentrations in Figure 4A and 4B. Also, the authors should show why the authors select indicated H2O2 concentrations in Figure 4A and 4B. Why do the authors use the different H2O2 concentration in this experiment? The authors should explain the difference of the viability of the control experiments conducted in the presence of H2O2. The sentence of “GSeSeG even at the concentration of 25 µM improved the cell viability against the H2O2 stress” (line 204 to 205) is strange. The reviewer does not understand the sentences from line 208 to 209.
Response 5: The concentrations of GSeSeG and H2O2 were pre-adjusted so that the viability of the cells became around 50 % in the assay. The reason for the use of a little higher concentration of H2O2 (i.e., 1.2 mM) in Figure 4B is to observe the antioxidant effects of GSeSeG clearly. If the same concentration of H2O2 (i.e., 1 mM) was employed, the difference in the viability using variable concentrations of GSeSeG was difficult to observe due to the saturation: the viability became more than 80 % by the treatment with GSeSeG as seen in Figure 4A. There was an error in Figure 4B: the bar of 1.2 mM H2O2 should not extend to CN. The cell viability for CN (without the treatment with GSeSeG and H2O2) was about 100 %, while it was about 20 % under the treatment with H2O2 (1.2 mM). We apologize for the confusion due to this mistake. Now, the figure is corrected, and more quantitative discussion is given in the text, subsection 2.4. Effects of GSeSeG on cell viability against oxidative stress caused by H2O2, as suggested by the reviewer.
Comments 6: The authors should explain why H2O2 concentrations (0, 1.2, 1.5 mM) are selected. The reason in described in line 228 is not persuasive. Figure 5A and 5B indicate reduction of the cell viability depending on H2O2 concentration. Therefore, the reviewer does not understand why the authors describe “the addition of MG significantly decreased the cell viability irrespective of the concentration of H2O2” (line 231). The expression described in line 232 is strange. The authors should explain the experimental results shown in Figure 5, accurately and briefly.
Response 6: In the revised manuscript, Figure 5 is rebuilt using the same data as before and added with the results of the t-test analysis. According to the revisions made on the figure and the text, the effects of MG as well as those of H2O2 would be clear now. As mentioned in the text, the H2O2 concentration added on the third day was set higher than those investigated in Figure 4, because the cells were more densely grown in this experiment: a high ratio of the viability (ca. 80 %) was obtained under the conditions of GSeSeG ‒, MG ‒, and 1.2 mM H2O2 in Figure 5. Therefore, we applied higher concentrations of H2O2 than that applied in Figure 4. The expression, “the addition of MG significantly decreased the cell viability irrespective of the concentration of H2O2”, was corrected as “the addition of MG significantly decreased the cell viability, depending on the concentration of H2O2”.
Comments 7: Subchapters in Discussion is unnecessary. If the authors would like to separate the contents, the authors should use the paragraph. The reviewer does not understand the contents of “3.3. Low cytotoxicity of GSeSeG”. The reviewer thinks that the contents make it impossible to discuss the low toxicity on GSeSeG.
Response 7: Thanks for the suggestion. We merged Discussion into Results now. As for the low toxicity of GSeSeG, we inserted the sentence, “This feature of GSeSeG was remarkable as compared with the cytotoxicity of selenite (Na2SeO3), for which the LB50 value was 1.2 μM against HeLa cells [34]”, citing one new reference, in subsection, 2.4. Effects of GSeSeG on cell viability against oxidative stress caused by H2O2. Compared to the cytotoxicity of selenite, it is obvious that GSeSeG has a low cytotoxicity.
Comments 8: Figure 6 should be rebuilt. “GSeSeG” (placed to the outer side of the cell) should be linked to “GSeSeG” in the cell, and “GSe-X” (placed to the outer side of the cell) should be linked to “GSe-X” in the cell.
Response 8: We corrected Figure 6 according to the suggestion. It is indeed easier to understand now.
Comments 9: Comments on the Quality of English Language
Moderate English proofreading is required.
Response 9: We made our best efforts to correct grammatical and typographical errors.
Reviewer 3 Report
Comments and Suggestions for Authors
The manuscript by Kanamori et al. entitled “Antioxidative and antiglycative stress activities of selenoglutathione diselenide” investigates the antioxidant properties of selenoglutathione diselenide. The manuscript is well-written and provides evidence that the investigated compound inhibits oxidative stress and glycative stresses in cell-free and cell culture-based models. However, some issues should be clarified:
Major issues:
- Why Hela cell line was chosen? The authors should explain why this particular cell line has been selected for this study. Why cancer cell line was used? Why normal cell lines were not used to compare the effects?
- Statistical analysis was not performed. The authors should statistically process the data and indicate in all the Figures whether the difference in parameters is statistically significant. Furthermore, the applied statistical tests should be mentioned in each Figure legend. Since statistical analysis is not presented, it is not clear which parameters are shown as columns and whiskers in Figures.
- The Discussion section should not contain subheadings and should focus on the advantages of the modified compound compared with the endogenous GSH and in which conditions it might be applied
- Limitations of the study should be acknowledged.
- The last paragraph of the Conclusion section is rather suitable for the Discussion section.
Minor issues:
- Lines 21-22: advanced glycation end-product
- Line 44: the same
- Line 399: subconfluent
- Which microplate reader was used for the neutral red assay
Author Response
Comments 1: The manuscript by Kanamori et al. entitled “Antioxidative and antiglycative stress activities of selenoglutathione diselenide” investigates the antioxidant properties of selenoglutathione diselenide. The manuscript is well-written and provides evidence that the investigated compound inhibits oxidative stress and glycative stresses in cell-free and cell culture-based models. However, some issues should be clarified:
Response 1: Thanks for the interest and valuable comments.
Major issues:
Comments 2: Why Hela cell line was chosen? The authors should explain why this particular cell line has been selected for this study. Why cancer cell line was used? Why normal cell lines were not used to compare the effects?
Response 2: This study is our first trial to investigate the biological activities of GSeSeG using a cell line. In this regard, we selected HeLa cells because it is widely used in biological research and easy to manipulate. This is now indicated in the subsection, 2.4. Effects of GSeSeG on cell viability against oxidative stress caused by H2O2 (l. 208-209). We will use normal cell lines in our future works as suggested.
Comments 3: Statistical analysis was not performed. The authors should statistically process the data and indicate in all the Figures whether the difference in parameters is statistically significant. Furthermore, the applied statistical tests should be mentioned in each Figure legend. Since statistical analysis is not presented, it is not clear which parameters are shown as columns and whiskers in Figures.
Response 3: This is the same as the comment 3 of Reviewer 1. We analyzed the p values for all the data shown in Figures 1 to 5, and the results are now included in Figures 1, 4, and 5. However, it seems to be trivial to show p values for Figures 2 and 3. Therefore, the p values are not included for these figures.
Comments 4: The Discussion section should not contain subheadings and should focus on the advantages of the modified compound compared with the endogenous GSH and in which conditions it might be applied
Response 4: In the revised manuscript, we merged the Discussion section to the Results section. By doing this, the advantages of GSeSeG over endogenous GSH become clearer now.
Comments 5: Limitations of the study should be acknowledged.
Response 5: We added the disadvantage of GSeSeG for the application to a clinical use in the last sentence of Conclusion.
Comments 6: The last paragraph of the Conclusion section is rather suitable for the Discussion section.
Response 6: Thanks for this suggestion. We divided the corresponding part and moved them into the Introduction and Results and Discussion
Comments 7: Minor issues:
Lines 21-22: advanced glycation end-product
Line 44: the same
Line 399: subconfluent
Response 7: We corrected accordingly.
Comments 8: Which microplate reader was used for the neutral red assay
Response 8: We now show the name of the microplate reader, which was used in this study.
Round 2
Reviewer 2 Report
Comments and Suggestions for Authors
The submitted revised manuscript describes the high antioxidant capacity of selenoglutathione (GSeH) and the investigation of GSeH’s effects on detoxification of the stress-causing substances using cell-free systems. Although the manuscript has been revised, the explanation of experimental results has not been improved. In addition, the reviewer still cannot understand the setting of concentrations that the authors have designed in each experiment. Therefore, if the authors do not present clearly a description of experiments and a convincing explanation on the concentrations of materials (reagents), the reviewer judges that the revised submitted manuscript would not be suitable for publication.
On Figure 1, it is too hard to read the explanation that the authors describe in the main text. Since Figure 2A is constructed based on the data shown in Figure 2B, Figure 2B is not required. The explanation of data shown in Figure 3 is still not good. The reviewer cannot understand the setting of GSeSeG concentration in Figure 3A. What is the peak at 9 min in Figure 3B? There is no specific explanation on this peak, although the authors explain the peaks in Figure 3C. The reviewer cannot understand the change of the cell viability by the slight difference of H2O2 concentration in Figure 4. Also, the reviewer cannot understand that the authors present both data with treatment with H2O2 (1.2 mM and 1.5 mM) in Figure 5. One is enough. Totally, the reviewer thinks that the authors do not present experiment results accurately.
Comments on the Quality of English LanguageModerate English proofreading is still required.
Author Response
General Comments: The submitted revised manuscript describes the high antioxidant capacity of selenoglutathione (GSeH) and the investigation of GSeH’s effects on detoxification of the stress-causing substances using cell-free systems. Although the manuscript has been revised, the explanation of experimental results has not been improved. In addition, the reviewer still cannot understand the setting of concentrations that the authors have designed in each experiment. Therefore, if the authors do not present clearly a description of experiments and a convincing explanation on the concentrations of materials (reagents), the reviewer judges that the revised submitted manuscript would not be suitable for publication.
Our response: Thank you for critical reading of the manuscript. We agree all the points and have revised the manuscript accordingly. All the data shown in figures were re-analyzed by one-way ANOVA and the host-hoc Newman-Keuls test as well as the t-test. The results are now included in the figures and the legends. We hope all the concerns are cleared now.
Comment 1: On Figure 1, it is too hard to read the explanation that the authors describe in the main text.
Response: According to this comment, we re-examined the paragraph after Figure 1 and have modified the expression of the results and discussion as highlighted in green. Along with the statistical analysis added to Figure 1, the explanation here would have been improved significantly.
Comment 2: Since Figure 2A is constructed based on the data shown in Figure 2B, Figure 2B is not required.
Response: The kinetic data that had been shown in old Figure 2A were obtained by fluorescence spectra of the solution, not by the HPLC data of old Figure 2B. However, we agree that old Figure 2B was redundant and would make the readers confused. Therefore, in new Figure 2, the HPLC data were deleted.
Comment 3: The explanation of data shown in Figure 3 is still not good. The reviewer cannot understand the setting of GSeSeG concentration in Figure 3A. What is the peak at 9 min in Figure 3B? There is no specific explanation on this peak, although the authors explain the peaks in Figure 3C.
Response: The GSeSeG concentration was first set to the same concentration as that applied in Figure 2 (i.e., 0.5 mM). Then, it was decreased to 0.05 mM gradually. This is now described explicitly in line 180-184 as highlighted in green. Yes, the peak at 9 min in Figure 3B was the same GSH-CDNB conjugate as observed in Figure 3C. This is now clearly indicated in Figure 3B as well as line 188-193.
Comment 4: The reviewer cannot understand the change of the cell viability by the slight difference of H2O2 concentration in Figure 4.
Response: The reason for the slight increase in the concentration of H2O2 in Figure 4B, i.e., from 1 mM to 1.2 mM, was because the cell viability against 1 mM H2O2 stress was a little too high (more than 80 %) to analyze the concentration-dependent effects of GSeSeG. This is now described in line 241-246.
Comment 5: Also, the reviewer cannot understand that the authors present both data with treatment with H2O2 (1.2 mM and 1.5 mM) in Figure 5. One is enough.
Response: We agree with this comment. So, the data in the presence of 1.5 mM H2O2 were deleted.
Comments on the Quality of English Language: Moderate English proofreading is still required.
Response: We scrutinized the manuscript carefully and tried to improve English expression as much as possible.
Reviewer 3 Report
Comments and Suggestions for Authors
The authors have addressed most comments. However, it is still not sufficient. The authors should add a separate subsection to the Materials and Methods section describing the statistical analyses performed. It is still not clear which parameters are shown as columns and whiskers (e.g., Mean ± SEM or Mean ± SD). This should be mentioned in the legends to Figures. Figure 2 has no indication of the statistical significance. The same is applied to Figure 3a. These results are not valid without proper statistical analysis. Figure 4b: Was the difference statistically significant only for 0.5 µM? Are higher concentrations not protective?
Author Response
General Comments: The authors have addressed most comments. However, it is still not sufficient.
Response: Thank you for careful reading of the revised manuscript. According to the comments, we re-revised the manuscript.
Comment 1: The authors should add a separate subsection to the Materials and Methods section describing the statistical analyses performed.
Response: We redone the statistical analysis of all the data with more appropriate method, i.e., one-way ANOVA and the host-hoc Newman-Keuls test, in addition to the t-test. According to the comment, a new subsection regarding to the statistical analysis is now added to the Materials and Methods section. See line 460-466.
Comment 2: It is still not clear which parameters are shown as columns and whiskers (e.g., Mean ± SEM or Mean ± SD). This should be mentioned in the legends to Figures.
Response: The columns and whiskers represent Mean ± SD. This is now clearly shown in the legends.
Comment 3: Figure 2 has no indication of the statistical significance. The same is applied to Figure 3a. These results are not valid without proper statistical analysis.
Response: We have carried out one-way ANOVA and the host-hoc Newman-Keuls test for the data shown in Figures 2 and 3A, and the results are now shown in the figures.
Comment 4: Figure 4b: Was the difference statistically significant only for 0.5 µM? Are higher concentrations not protective?
Response: According the post-hoc test, all data for GSeSeG (from 0.5 to 25 µM) are not of significant difference. This is now clearly shown in Figure 4B.
Round 3
Reviewer 2 Report
Comments and Suggestions for Authors
The submitted revised manuscript describes the high antioxidant capacity of selenoglutathione (GSeH) and the investigation of GSeH’s effects on detoxification of the stress-causing substances using cell-free systems. Although the manuscript has been revised again, the explanation of experimental results has not been improved. Especially, the presentation of all graphs shown in Figure has gotten worse. The reviewer cannot understand the intension that the authors use the alphabet (A, a, and others) in the graph. Therefore, the reviewer recommends resubmission of the manuscript after the authors reconstruct the contents of the manuscript and rewrite a manuscript.
Comments on the Quality of English LanguageMinor English proofreading is required.
Author Response
Comments and Suggestions for Authors:
The submitted revised manuscript describes the high antioxidant capacity of selenoglutathione (GSeH) and the investigation of GSeH’s effects on detoxification of the stress-causing substances using cell-free systems. Although the manuscript has been revised again, the explanation of experimental results has not been improved. Especially, the presentation of all graphs shown in Figure has gotten worse. The reviewer cannot understand the intension that the authors use the alphabet (A, a, and others) in the graph. Therefore, the reviewer recommends resubmission of the manuscript after the authors reconstruct the contents of the manuscript and rewrite a manuscript.
Response:
Thank you for your comment. We have applied one-way ANOVA/Newman-Keuls for comparison of three or more groups and t-test for comparison of two groups. To represent the differences observed in the Newman-Keuls test after one-way ANOVA, we used letters. Bars sharing the same letter are not significantly different according to one-way ANOVA and the post-hoc Newman-Keuls test, while different letters represent a statistical significance (p < 0.05). In some graphs, the results are from two different one-way ANOVA. In these cases, . We added this information that was missing in the legends. In contrast, to represent the differences between only two groups assessed through t-test, we used the asterisks. For clarity, we use * p<0.05, **P<0.01, and *** p<0.001. We used both letters and asterisks to differentiate one-way ANOVA/Newman-Keuls test and t-test, respectively. The use of letters to demonstrate the statistical significances has been largely applied in the literature. We believe that this is one of the best ways to clearly express the analysis of statistical analysis in scientific papers. See the following paper, for example: Pharmaceuticals 2024, 17, 603 (https://doi.org/10.3390/ph17050603).
Comments on the Quality of English Language:
Minor English proofreading is required.
Response:
We tried to fix the wrong expressions and typographical errors as much as possible.
Reviewer 3 Report
Comments and Suggestions for Authors
The authors have improved the manuscript by addressing the comments.
Minor issues:
- Add software used to statistically process the data to the corresponding subsection of the Materials and Methods section
Author Response
Comments and Suggestions for Authors:
The authors have improved the manuscript by addressing the comments.
Response:
Thank you for understanding our revisions to improve the manuscript.
Minor issues:
Add software used to statistically process the data to the corresponding subsection of the Materials and Methods section
Response:
All statistical analysis was carried out by Microsoft Excel program. This is now indicated in the corresponding section as shown below.
“All statistical analyses were carried out by Microsoft Excel program using built-in functions for one-way ANOVA and t-test. For the post-hoc test, the analysis was manually done using the same program.”
Round 4
Reviewer 2 Report
Comments and Suggestions for Authors
The submitted revised manuscript describes the high antioxidant capacity of selenoglutathione (GSeH) and the investigation of GSeH’s effects on detoxification of the stress-causing substances using cell-free systems. Although the manuscript has been revised again, the explanation of experimental results has not been improved. The reviewer does not doubt the reliability of the experimental data, but points out that the explanation of the experimental results is strange and should be clearer. It is too hard to understand the description of the experimental results by the authors. For instance, the sentence of “In the presence of GPx (+GPx), the reduction of H2O2 with GSH should be catalyzed by the enzyme because the H2O2 reduction rate became low in the absence of GPx, where H2O2 should be reduced through a direct reaction with GSH.” (lane 119 to 121) should be much clearer. One sentence contains both “In the presence of GPx” and “in the absence of GPx”. No one requires excessive explanations (such as protocols) in figure legend (Figure 1). And, as the reviewer has pointed out repeatedly, the reagent concentrations that the authors have set are strange. The authors used 1.0 mM of GSeH in the experiments shown in Figure 1, but used 0.5 mM of GSeH in Figure 2. The authors used 0.5 mM of GSeSeG at the maximum in Figure 3, but used 25 mM (0.025 mM) of GSeSeG at the maximum in Figure 4, and 3.0 mM (0.003 mM) of GSeSeG at the maximum in Figure 5. If these data are correct, a reducing ability of GSeH (GSeSeG) would be exerted at a lower concentration than GSH (GSSG).
Anyway, the reviewer recommends resubmission of the manuscript after the authors reconstruct the contents of the manuscript and rewrite a manuscript.
Comments on the Quality of English Language
Moderate English proofreading is required.